# Association of *CYP2C19, CYP2D6* and *CYP3A4* Genetic Variants on Primaquine Hemolysis in G6PD-Deficient Patients

**DOI:** 10.3390/pathogens12070895

**Published:** 2023-06-30

**Authors:** Marielle M. Macêdo, Anne C. G. Almeida, Gabrielly S. Silva, Amanda C. Oliveira, Victor I. Mwangi, Ana C. Shuan, Laila R. A. Barbosa, Fernanda Rodrigues-Soares, Gisely C. Melo

**Affiliations:** 1Programa de Pós-graduação em Ciências Aplicadas à Hematologia, Universidade do Estado do Amazonas, Manaus 69040-000, AM, Brazil; 2Fundação de Medicina Tropical Dr. Heitor Vieira Dourado, Manaus 69040-000, AM, Brazil; 3Programa de Pós-graduação em Medicina Tropical, Universidade do Estado do Amazonas, Manaus 69040-000, AM, Brazil; 4Departamento de Patologia, Genética e Evolução, Instituto de Ciências Biológicas e Naturais, Universidade Federal do Triângulo Mineiro, Uberaba 35025-250, MG, Brazil

**Keywords:** *Plasmodium vivax*, malaria, primaquine hemolysis, G6PD deficiency, *CYP2C19*, *CYP2D6*, *CYP3A4*

## Abstract

In the Amazon, the treatment for *Plasmodium vivax* is chloroquine plus primaquine. However, this regimen is limited due to the risk of acute hemolytic anemia in glucose-6-phosphate dehydrogenase deficiency. Primaquine is a prodrug that requires conversion by the CYP2D6 enzyme to be effective against malaria. A series of cases were performed at an infectious diseases reference hospital in the Western Brazilian Amazon. The STANDARD G6PD (SD Biosensor^®^) assay was used to infer G6PD status and real-time PCR to genotype *G6PD, CYP2C19*, *CYP2D6* and CYP3A4. Eighteen patients were included, of which 55.6% had African A− variant (G202A/A376G), 11.1% African A+ variant (A376G), 5.6% Mediterranean variant (C563T) and 27.8% were wild type. CYP2C19, CYP2D6 and CYP3A4 genotyping showed no statistically significant differences in the frequency of star alleles between the groups G6PD deficient and G6PD normal. Elevated levels of liver and kidney markers in the G6PDd patients were observed in gNM, gRM and gUM of CYP2C19 and CYP2D6 (*p* < 0.05). Furthermore, in this study there was no influence of CYPs on hemolysis. These findings reinforce the importance of studies on the mapping of G6PD deficiency and genetic variations of CYP2C19, CYP2D6 and CYP3A4. This mapping will allow us to validate the prevalence of CYPs and determine their influence on hemolysis in patients with malaria, helping to decide on the treatment regimen.

## 1. Introduction

In Amazon, *Plasmodium vivax* is the most prevalent species to cause malarial infections, which produces dormant forms of the parasite in the liver (hypnozoites) that can cause relapses weeks or months after an infection, thus causing ongoing transmission [1,2]. The elimination of hypnozoites requires an 8-aminoquinoline such as primaquine (PQ) or tafenoquine (TQ). However, the use of these drugs is limited by the risk of acute hemolytic anemia (AHA) in individuals with glucose-6-phosphate dehydrogenase deficiency (G6PDd) [3,4,5,6], which can be potentially fatal [7]. Hemolytic crises are often reported to induce jaundice, abdominal pain, fever, dark urine, moderate to severe anemia, increased serum levels of lactate dehydrogenase (LDH) and indirect bilirubin and low or undetectable haptoglobin [3]. 

Glucose-6-phosphate dehydrogenase (G6PD) is an enzyme involved in protecting erythrocytes against oxidative stress [1,4]. This erythrocytic enzymopathy has an X-linked recessive pattern and is associated with individuals of Mediterranean, Asian and African descent [3,5,6]. In the Brazilian population, studies have reported a frequency of 4.5 to 10% of G6PDd [8,9], and G6PDd variants include the African, Mediterranean, Seattle, Chatham, Bahia and São Paulo alleles [6,10,11,12]. However, the African variant is the most common and is characterized as A− (G202A/A376G) [13,14,15], and in Brazil this variant is also the most common [16].

Previously, in the Amazon region, cases of PQ-induced hemolysis have been reported in G6PDd individuals with severe anemia, acute kidney injury (AKI) and blood transfusion [7,8]. However, little is known about the influence of genetic alterations on cytochrome P450 (CYP) and clinical complications of PQ therapy in this population.

PQ is a prodrug whose therapeutic efficacy of PQ appears to be strongly correlated with the activity of the highly polymorphic metabolic enzyme cytochrome P450 2D6 (CYP2D6) [5,17,18]. *CYP2D6**1 and *CYP2D6**2 variants have been reported as the most common in Brazil, with the frequency of individuals with more than these two active copies ranging from 6.3% in southern Brazil to 10.2% in northern Brazil, which can accelerate drug metabolism [19]. Other pharmacogenes, such as *CYP2C19* and *CYP3A4*, have also been studied in the therapeutic response of PQ in *P. vivax*-infected patients [20,21]. Despite studies reporting severe PQ-induced hemolysis in G6PDd people, there is a lack of information about the role of CYP genetic variants and the clinical complications of PQ therapy in this population [16]. In this context, considering the diverse functional properties of CYPs, and the scarcity of studies about PQ metabolism associated with G6PD deficiency, this study aimed to evaluate the association of CYP2C19, CYP2D6 and CYP3A4 genetic variants with primaquine hemolysis in G6PD-deficient patients.

## 2. Materials and Methods

Medical records from patients admitted to Fundação de Medicina Tropical Dr Heitor Vieira Dourado (FMT-HVD) from 2015 to 2021 were screened for eligibility for this study, for a data survey of retrospective hospitalized cases. The FMT-HVD is a tertiary reference health care center for infectious diseases in Manaus, Western Brazilian Amazon, and receives patients directly seeking care and those referred by public and private care networks in Manaus and the surrounding cities. 

### 2.1. Study Subjects

Inclusion criteria were either gender, >6 months old, hospitalized, diagnosed with vivax malaria, treated with PQ and the presence of hemolysis. Non-inclusion criteria were unavailability of telephone contact, not residing in the city of Manaus or unable to participate. After telephone contact and a successful inclusion process, whole blood was collected to determinate G6PD status and carry out genotyping tests. Clinical and laboratory retrospective information was obtained from the electronic medical record, such as length of hospitalization, degree of anemia and jaundice, complete blood count, total bilirubin and fractions, oxaloacetic transaminase, pyruvic transaminase, lactate dehydrogenase, glucose, creatinine, urea, hemoglobinuria, proteinuria, bilirubinuria, need for blood transfusion, acute kidney injury and dialysis. According to the Brazilian malaria treatment guide, standard treatment was with CQ for 3 days (10 mg/kg on day 1 and 7.5 mg/kg on days 2 and 3) and with PQ at a dose of 0.5 mg/kg/day for 7 days, including children from 1 year old. For G6PDd, the therapeutic regimen contains a weekly dose of PQ for 8 weeks, adjusted by weight (0.75 mg/kg/day) and should be started on day 4 after treatment with CQ; children from 1 year are treated following the same guideline [22]. Anemia and jaundice were categorized using crosses according to intensity: +/2, +, ++ or + + +. The severity of anemia was classified according to the recommendations of the World Health Organization (WHO) [23].

### 2.2. G6PD Phenotyping

Hemoglobin concentration and G6PD enzyme activity were determined using a STANDARD G6PD biosensor kit (SD BIOSENSOR^®^, Yeongtong-gu, Suwon-si, Gyeonggi-do, Republic of Korea). The assay was performed according to the manufacturer’s instructions. The G6PD status classification considered, for women, ≤3.9 IU/gHb as deficient, 4.0–6.0 IU/gHb as intermediate and ≥6.0 IU/gHb as normal; for men ≤ 3.9 IU/gHb was considered deficient and ≥4.0 IU/gHb normal [24,25]. Participants were categorized into two groups: G6PDd (patients with G6PD deficiency or intermediate activity of the enzyme) and G6PDn (non-deficient of G6PD).

### 2.3. Genotyping of G6PD and CYPs

Genomic DNA was extracted using the QIAmp^®^ Blood Mini kit (Qiagen, Hilden, Germany), following the manufacturer’s recommendations. The targeted *G6PD* variants were: African A− G202A/A376G (rs1050828 and rs1050829), African A+ only A376G (rs1050829) and Mediterranean C563T (rs5030868). Eleven variants were selected for *CYP2D6*: 2549delA (rs35742686), 100C>T (rs1065852), 1846G>A (rs3892097), 4180G>C (rs1135840), 2988G>A (rs28371725), 3183G>A (rs59421388), 1584C>G (rs1080985), 1023C>T (rs28371706), 2615_2617delAAG (rs5030656), 31G>A (rs769258), 2850C>T (rs16947). Three variants were selected in *CYP2C19*: 681G>A (rs4244285, *2), 636G>A (rs4986893, *3), −806C>T (rs12248560, *17); and one variant in *CYP3A4*: −392A>G (rs2740574, *1B). G6PD, *CYP2C19*, *CYP2D6* and *CYP3A4* genotyping tests were performed using real-time PCR (7500 Fast Real-Time PCR System), Applied Biosystems, Foster City, CA, USA and TaqMan^®^ probes, using TaqMan™ Drug Metabolism Genotyping Assay (ThermoFisher scientific^®^, South San Francisco, CA, USA), following manufacturer instructions. *CYP* diplotypes were inferred using the HaploStats package implemented on the R platform and the identification of star (*) alleles followed the current nomenclature in the Pharmacogene Variation (PharmVar) Consortium [21,26,27]. The activity score (AS) system was used to determine the predicted CYP2D6 phenotype [28]. 

The determination of the number of copies (CNV) of the CYP2D6 gene was performed by real-time duplex PCR, following manufacturer instructions. The reference gene used in the assay was RNAseP. The CNV assay was performed using a 7500 Fast Real-Time PCR System, and the amplification product was analyzed using CopyCaller v2.1 software (Applied Biosystems, Foster City, CA, USA).

### 2.4. Ethical Considerations

The Ethics Review Board of the FMT-HVD approved the study (CAAE: 39179720.9.0000.0005). Subjects were informed of the study objectives and gave their consent for participation via a signed informed consent form before participating in the study. 

### 2.5. Statistical Analysis

Microsoft^®^ Excel^®^ 2019 was used for database management. Before performing the *t*-test or the Wilcoxon–Mann–Whitney test for the mean and standard deviation (SD) or median and interquartile range (IQR) values, continuous variables were subjected to the Shapiro–Wilk test for normality of the distribution. Categorical variables were expressed in absolute value (n) and relative frequency (%) before the frequency distribution was tested for significant difference using chi-square or Fisher’s exact tests. Where applicable, ANOVA or Kruskal–Wallis tests were performed to compare the clinical profile of the PQ-treated patients and their *CYP2C19* and *CYP2D6* phenotype profiles. In the analyses, patients with intermediate G6PD activity were defined in the same group as deficient patients (G6PDd). Results were considered statistically significant at *p* < 0.05. Analyses were performed using STATA (v13.1, StataCorp, College Station, TX, USA) and GraphPad Prism (v9.0.2) software.

## 3. Results

### 3.1. Demographic Characteristics

During the study period, 406 patients were diagnosed with vivax malaria and admitted to FMT-HVD, according to electronic medical records. Of these, 252 had no history of hemolysis on admission and 126 had no telephone contact. Twenty-eight patients fulfilled the inclusion criteria and with whom telephone contacts were made. Ten patients resided outside Manaus or were unable to participate, thus resulting in eighteen included participants (Figure 1).

The mean age of the participants was 25.4 years (G6PDd: 22.1 years and G6PDn: 34 years; *p* = 0.2269). The minimum age presented by the included participants was 2 years and the maximum age was 60 years. There was a general predominance of males (72.2%) in both groups, although gender distribution based on G6PD status was not statistically significant (*p* = 0.567). The sociodemographic data are described in Table 1. After the enzymatic measurement of G6PD, 13 (72.2%) were G6PDd (median 2.5 IU/g) and 5 (27.8%) were G6PDn (median 8.4 IU/g) (*p* = 0.0013).

### 3.2. Laboratory Characteristics

A hematological analysis demonstrated values below normal for both groups but with no statistical difference for erythrocytes, hematocrit (Ht), hemoglobin (Hb) and thrombocytopenia (*p* > 0.05). In the G6PDd group, there was a higher frequency of anemia and jaundice with no significant statistical difference. Biochemical markers of hepatic and renal injury in the two groups had increased levels, but with no significant statistical difference (Table 1). Furthermore, hemoglobinuria and proteinuria were observed in both groups. Red blood cell concentrate (RBCC) transfusion, acute kidney injury (AKI) and dialysis were reported only in the G6PDd group (Table 1). 

When analyzing the G6PD genotyping, it was observed that the African A− variant (G202A/A376G) was the most common with 55.6%, followed by African A+ (A376G) and the Mediterranean variants (C563T), with frequencies of 11.1% and 5.6%, respectively. Five (27.8%) participants were subsequently classified as wild type (Figure 2 and Table 2). 

Genotyping of *CYP2C19*, *CYP2D6* and *CYP3A4* showed no significant difference in allele frequency between groups (*p* > 0.05). The *CYP2C19**1 allele (wild-type variant) was the most frequent in this population (G6PDd: 73.1%; G6PDn: 90%; *p* = 0.269). *CYP2C19**2 (null function) was present only in the G6PDd group (15.4%) (*p* = 0.254), and *CYP2C19**17 (increased function) had a frequency of 11.5% and 10.0% (*p* = 0.695) in G6PDd and G6PDn, respectively (Table 2). The *CYP2C19**3 allele was absent in this population (Table 3). The frequency of predicted *CYP2C19* phenotypes did not present significant differences between groups (*p* > 0.05). The normal metabolizer (gNM) had a frequency of 46.1% in G6PDd and 80.0% in G6PDn (*p* = 0.225) patients, and the rapid metabolizer (gRM) had frequencies of 23.1% and 20.0% (*p* = 0.701) in G6PDd and G6PDn patients, respectively. The intermediate metabolizer (gIM) was observed only in the G6PDd group (30.8%) (*p* = 0.234).

In the *CYP2D6* gene, the frequencies of null function (*4, *5), decreased function (*17, *29) and normal function (*1, *2, *34, *35) alleles were similar between groups (*p* > 0.05). *CYP2D6**2 was present only in G6PDd group (53.8%) (*p* = 0.054). The CNV analysis identified two (11.1%) individuals with multiplications in *CYP2D6,* but one of these individuals was heterozygous and it was not possible to identify the multiplied allele (Table 3). Ultra-rapid allele frequency (*1xN) was 7.7% and observed in the G6PDd group (*p* = 0.722). This individual with the ultra-rapid allele had severe hemolysis, severe anemia and elevated liver and kidney markers, such as BT median 8.8 mg/dL, BD median 5.3 mg/dL, BI median 5.3 mg/dL, SGOT median 99 U/L, SGPT median 154 U/L, LDH median 2044 U/L, glucose mean 144 mg/dL, creatinine median 9.7 mg/dL and urea median 277 mg/dL (Appendix A). The predicted *CYP2D6* phenotype was significant for gIM, with frequencies of 7.7% and 60.0% (*p* = 0.044) in the G6PDd and G6PDn groups, respectively. The gNM phenotype had a frequency of 84.6% in G6PDd and 40.0% in G6PDn (*p* = 0.099) groups, and the ultra-rapid metabolizer (gUM) was found only in G6PDd (7.7%) (*p* = 0.722) patients (Table 2).

Regarding the frequency of *CYP3A4*, the *1 allele (normal function) was more frequent in both groups (G6PDd: 92.3%; G6PDn: 70.0%; *p* = 0.119), and *CYP3A4**1B had a frequency of 7.7% and 30.0% in G6PDd and G6PDn (*p* = 0.119), respectively (Table 2). 

Comparisons of the clinical and laboratory profile in G6PDd patients and *CYP2C19*, *CYP2D6* and *CYP3A4* phenotypes showed a statistically significant difference between total bilirubin and fractions, SGPT and urea (*p* < 0.05) (Appendix A). The *CYP3A4* *1/*1 genotype was present in G6PDd patients with variants G202A/A376G and C563T (90.1% and 9.1%; *p* = 0.013), respectively. The *1/*1B genotype was present only in individuals with the A376G variant (100.0%; *p* = 0.013) (Appendix A). Among the G6PDd individuals with CYP2C19 gRM, a higher TB median was observed (gNM: 0.5 mg/dL; gIM: 0.3 mg/dL; gRM: 1.3 mg/dL; *p* = 0.0494). The same was observed in SGPT (gNM: median 57.5 U/L; gIM: median 39 U/L; gRM: median 154 U/L; *p* = 0.0052) (Appendix A). The G6PDd patient with CYP2D6 gUM had higher serum levels of TB (gNM: median 2.3 mg/dL; gIM: median 0.8 mg/dL; gUM: median 8.8 mg/dL; *p* = 0.0048), DB (gNM: median 0.5 mg/dL; gIM: 0.3 mg/dL; gUM: median 5.3 mg/dL; *p* < 0.0001) and IB (gNM: median 1.6 mg/dL; gIM: median 0.5 mg/dL; gUM: median 5.3 mg/dL; *p* = 0.0392). Increased median levels of urea were observed in subjects with 2D6 gUM (gNM: 43 mg/dL; gIM: 10 mg/dL; gUM: 277 mg/dL; *p* = 0.0256). 

Comparisons of the clinical and laboratory profile in G6PDn did not show statistically significant differences between phenotypes of CYP2C19, CYP2D6 and *CYP3A4* genotypes (*p* > 0.05) (Appendix A).

## 4. Discussion

The antimalarials PQ and TQ are effective drugs that eliminate the hepatic stages of *P. vivax* and prevent relapses. However, this class of 8-aminoquinolines can cause hemolysis in individuals with low G6PD enzyme activity, which presents an obstacle to the global implementation of drug delivery against vivax malaria [29,30]. The prevalence of G6PDd in the male population has been reported in several studies from around the world, and also in the Amazon region, with a frequency of 4.5% to 10% [14,16,31]. In our study, the African A− variant (G202A/A376G) was the most common, around 55.6%, followed by the African A+ (A376G) (11.1%) and Mediterranean (C563T) (5.6%) variants. The phenotypic diagnosis in heterozygous women remains a challenge, since the random inactivation of the X chromosome can occur, resulting in mosaicism. Consequently, about half of the erythrocytes are normal and the other half are G6PDd [6,32]. Most of the women identified in our study were heterozygous for the African A− variant, which reinforces the enzymatic activity of G6PD at higher values, classifying them as having an intermediate deficiency. In contrast, one female patient who was heterozygous for the African A+ variant had the lowest enzymatic activity, which may have influenced the severity of hemolysis, along with the presence of CYP2C19 gRM. In a systematic review carried out by Rodrigues-Soares et al. (2018), the *CYP2C19**2 and *CYP2C19**17 alleles were found to be more frequent in Mixed patients in the Brazilian population, 14% and 19%, respectively [33] and, in our study, the frequency of these alleles was 11.1%. The *CYP2D6**1 allele, followed by *2, had the highest frequencies, 77.8% and 38.9%, respectively. In Brazil, the *1 and *2 alleles are the most frequent and are widely distributed in different geographic regions [17,19]. 

Considering the AHA parameters, G6PDd patients had the worst hemolytic condition during hospitalization, compared to G6PDn patients. Although there was no significant difference, they had a decrease in erythrocytes, hematocrit and hemoglobin, an increase in liver and kidney markers and a need for blood transfusion and dialysis. Therefore, these patients also had greater clinical and laboratory complications when compared to predicted CYP2C19 (gRM) and CYP2D6 (gUM) phenotypes and *CYP3A4* *1/*1B genotype. DB and SGPT showed statistical significance among the CYP2C19 phenotypes. Among the CYP2D6 phenotypes, there was significant significance in TB, DB, BI and urea. In *CYP3A4* genotypes, individuals with the *1/*1B genotype had higher levels of all markers, except glucose and creatinine (*p* > 0.05). These findings still do not allow us to affirm the influence of these genetic alterations on the hemolysis process, although fast and ultra-rapid metabolizers suffered a greater clinical impact.

Both *G6PD* and *CYP2D6* are highly polymorphic genes, which makes the use of PQ more complex [34]. In studies that evaluated the pharmacokinetics of PQ with CYP2D6 metabolizers, it was observed that drug concentration was higher in gPMs, and that the low production of active metabolites was responsible for its effectiveness, which would result in relapse and failure [17]. In gIMs, the active metabolites may not achieve the desired response, and the clinical picture would depend on the severity of the CYP2D6 enzymatic alteration, body weight and the total dose administered. A study with 88 patients infected with vivax malaria from Thailand identified that about 51.1% of these patients were intermediate metabolizers of CYP2D6 and had reduced enzymatic activity, thus putting them at risk of therapeutic failure and reinforcing the need to implement G6PD and CYP2D6 testing as an aid in the use of 8-aminoquinolines for the elimination of vivax malaria [15,35]. In gUM, the clinical outcome with PQ remains undetermined [17]; however, it is suggested that the CYP2D6 ultra-rapid phenotype is less susceptible to recurrence and PQ failure [21]. 

This study had some limitations, such as the low number of patients included, due to the scarcity of well-described severe hemolytic cases followed up with exams and updated phone contact record; clinical and laboratory data were collected retrospectively. The future implementation of pharmacogenetic testing for better patient management is needed, The implementation of pharmacogenetic tests would help in better patient management, as the metabolism of PQ by different enzymatic phenotypes of CYPs can contribute to severe hemolysis, thus increasing the risks of hospitalizations [5,21,34,35], in addition to the importance of diagnosing G6PDd before treatment with any of the 8-aminoquinolines [36,37].

## 5. Conclusions

In our study, from a malaria-endemic area in northern Brazil, the *G6PD* variant African A − (G202A/A376G) was predominant among individuals with low activity. The G6PDd individuals showed the worst hemolytic condition during hospitalization, and the comparison of the predicted phenotype of *CYP2C19* and *CYP2D6* with the clinical and laboratory profile of these individuals identified gRM and gUM as having the highest serum levels of hemolysis markers. The CYP3A4 *1/*1B genotype had the greatest clinical impact in both G6PD groups. These findings reinforce the importance of studies in a larger G6PD-deficient population and the association of the genetic variants of *CYP2C19, CYP2D6* and *CYP3A4* with the hemolytic process during malaria vivax infection in both G6PD groups.

## Figures and Tables

**Figure 1 pathogens-12-00895-f001:**
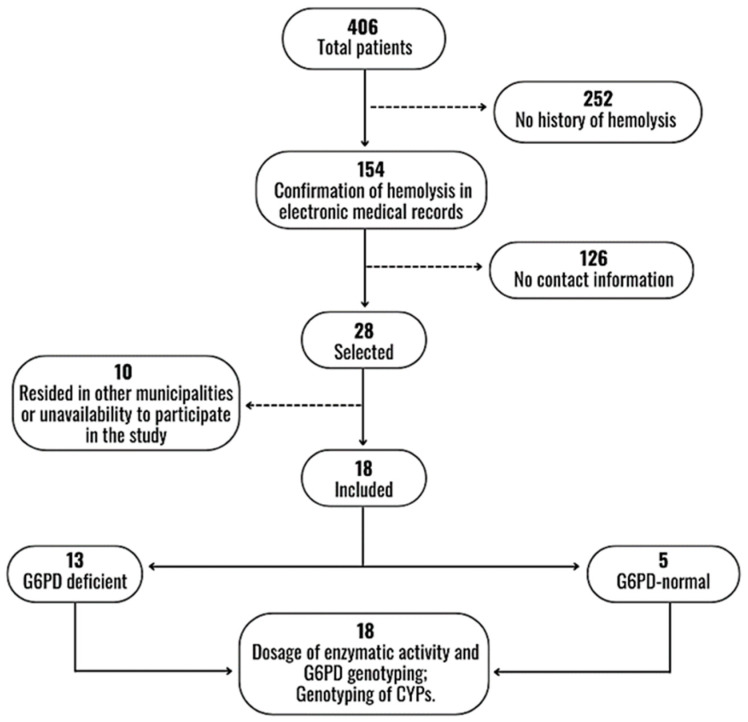
Flowchart of selection and inclusion of participants.

**Figure 2 pathogens-12-00895-f002:**
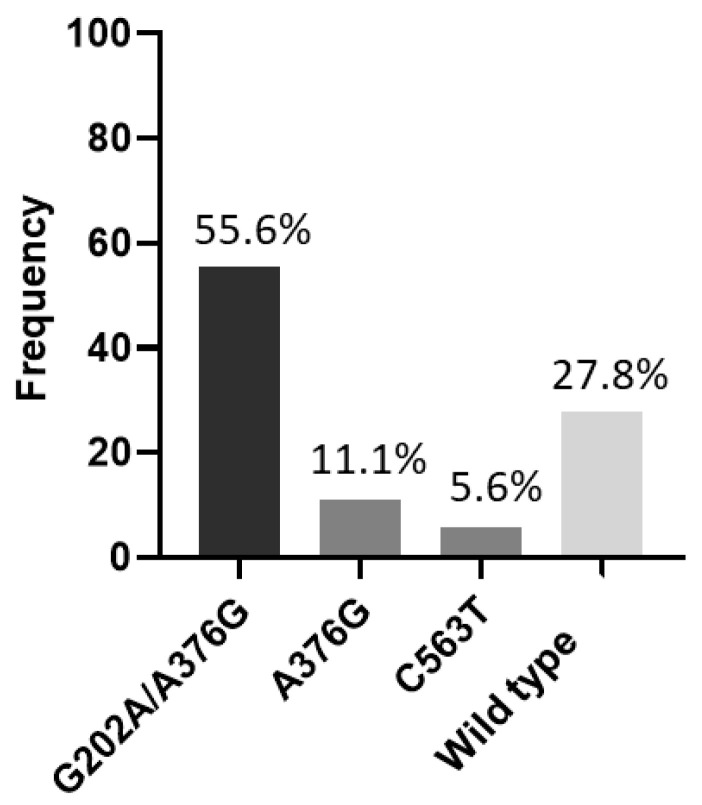
Frequency of G6PD variants. G202A/A376G = African A−; A376G = African A+; C563T = Mediterranean; Wild type = G6PDn.

**Table 1 pathogens-12-00895-t001:** Sociodemographic, clinical and laboratory profile in G6PDd and G6PDn treated with PQ for *P. vivax*.

	Total	G6PDd Median 2.5 IU/g Hb (IQR = 2–3.1)	G6PDn Median 8.4 IU/g Hb (IQR = 7.8–9.7)	*p* Value
Age, years, mean (SD)	25.4 (18.2)	22.1 (14.4)	34 (25.7)	0.2269
Sex				
Female, n/N (%)	5 (27.8)	4 (30.8)	1 (20.0)	0.567
Male, n/N (%)	13 (72.2)	9 (69.2)	4 (80.0)	
Hospitalization, days, median (IQR)	4 (3–7)	4 (3–14)	4 (4–5)	0.6175
Level of anemia, n/N (%)				
+/2	2 (13.3)	1 (8.3)	1 (33.3)	0.446
+	3 (20.0)	2 (16.7)	1 (33.3)	
++	8 (53.3)	7 (58.3)	1 (33.3)	
+++	2 (13.3)	2 (16.7)	-	
Level of jaundice, n/N (%)				
+/2	-	-	-	0.345
+	2 (18.2)	1 (11.1)	1 (50.0)	
++	9 (81.8)	8 (88.9)	1 (50.0)	
+++	-	-	-	
Erythrocyte (millions/ mm^3^), mean (SD)	2.7 (1.1)	2.6 (0.97)	3.6 (1.0)	0.0655
Ht (%), mean (SD)	24.3 (7.2)	22.4 (6.3)	29.1 (7.7)	0.0716
Hb (g/dL), mean (SD)	8.3 (2.1)	7.7 (1.8)	9.7 (2.4)	0.0661
Platelets (mm^3^), median (IQR)	81,420 (65,160–315,000)	194,000 (70,150–353,000)	71,000 (47,570–73,290)	0.1263
TB (mg/dL), median (IQR)	2.3 (1.15–4)	2.4 (1.15–4)	1.9 (1.2–7.2)	0.9516
DB (mg/dL), median (IQR)	0.5 (0.4–1.1)	0.5 (0.4–1.1)	0.8 (0.4–4)	0.6668
IB (mg/dL), median (IQR)	1.4 (0.8–3.2)	1.6 (0.8–3.2)	1.2 (0.8–3.2)	>0.999
SGOT (U/L), median (IQR)	58.5 (35–101)	62 (45–101)	35 (33–56)	0.2565
SGPT (U/L), median (IQR)	52 (36–66)	59 (46–66)	36 (30–37)	0.1674
LDH (U/L), median (IQR)	1337.5 (772–2119)	1424 (859–2422)	601 (414–1613)	0.1391
Glucose (mg/dL), mean (SD)	119.6 (28.6)	116.9 (28.6)	147 (-)	-
Creatinine (mg/dL), median (IQR)	0.95 (0.5–2.2)	1.0 (0.7–1.6)	0.5 (0.4–2.2)	0.3743
Urea (mg/dL), median (IQR)	42 (27–69)	43 (28–75)	36 (27–49)	0.5873
Hemoglobinuria, n/N (%)	9 (81.8)	8 (80.0)	1 (100.0)	0.818
Proteinuria, n/N (%)	7 (63.6)	6 (60.0)	1 (100.0)	0.636
Bilirubinuria, n/N (%)	3 (27.3)	3 (30.0)	0 (0.0)	0.727
Transfusion (RBCC), n/N (%)	9 (56.3)	9 (69.2)	0 (0.0)	0.062
AKI, n/N (%)	2 (12.5)	2 (15.4)	0 (0.0)	0.650
Dialysis, n/N (%)	2 (12.5)	2 (15.4)	0 (0.0)	0.650

Abbreviations: G6PDd = G6PD deficient, G6PDn = G6PD normal, Ht = hematocrit, Hb = hemoglobin, TB = total bilirubin, DB = direct bilirubin, IB = indirect bilirubin, SGOT = oxaloacetic transaminase, SGPT = pyruvic transaminase, LDH = lactic dehydrogenase, RBCC = red blood cell concentrate, AKI = acute kidney injury, SD = standard deviation, IQR = interquartile range.

**Table 2 pathogens-12-00895-t002:** Allele frequency of *CYP2C19, CYP2D6, CYP3A4 and G6PD* and predicted phenotype of *CYP2C19* and *CYP2D6*.

Gene	Allele	Total		G6PDd		G6PDn		*p* Value
		n	(%)	n	(%)	n	(%)	
** *CYP2C19* ** **Predicted *CYP2C19* phenotype**	*1	28	77.8	19	73.1	9	90.0	0.269
	*2	4	11.1	4	15.4	0	0.0	0.254
	*17	4	11.1	3	11.5	1	10.0	0.695
	gNM	10	55.6	6	46.1	4	80.0	0.225
	gIM	4	22.2	4	30.8	0	0.0	0.234
	gRM	4	22.2	3	23.1	1	20.0	0.701
** *CYP2D6* **	*1	14	77.8	11	84.6	3	60.0	0.299
	*2	7	38.9	7	53.8	0	0.0	0.054
	*4	3	16.7	1	7.7	2	40.0	0.172
	*5	1	5.6	0	0.0	1	20.0	0.278
	*17	3	16.7	2	15.4	1	20.0	0.650
	*29	1	5.6	1	7.7	0	0.0	0.722
	*34	5	27.8	3	23.1	2	40.0	0.433
	*35	1	5.6	0	0.0	1	20.0	0.278
**Predicted *CYP2D6* phenotype**	*1xN	1	5.6	1	7.7	0	0.0	0.722
	gNM	13	72.2	11	84.6	2	40.0	0.099
	gIM	4	22.2	1	7.7	3	60.0	0.044
	gUM	1	5.6	1	7.7	0	0.0	0.722
** *CYP3A4* **	*1	31	86.1	24	92.3	7	70.0	0.119
	*1B	5	13.9	2	7.7	3	30.0	0.119
	A376G	2	11.1	2	15.4	0	0.0	-
** *G6PD* **	G202A/A376G	10	55.6	10	76.9	0	0.0	-
	C563T	1	5.6	1	7.7	0	0.0	-
	Wild type	5	27.8	0	0.0	5	100.0	-

Abbreviations: G6PDd = G6PD deficient, G6PDn = G6PD normal, gNM = normal metabolizer, gIM = intermediate metabolizer, gRM = rapid metabolizer, gUM = ultra-rapid metabolizer, *1xN = number of copies, A376G = African variant A+, G202A/A376G = African variant A−, * refers to the star allele.

**Table 3 pathogens-12-00895-t003:** Genotyping of *CYP2C19, CYP2D6, CYP3A4* and *G6PD* in G6PDd and G6PDn.

ID	Variant G6PD	Dosage of G6PD (IU/g Hb)	*CYP2C19*		*CYP2D6*			*CYP3A4*
			Genotype	Phenotype	Genotype	Phenotype	CNV	Genotype
**HPQ01**	**G202A/A376G**	2.5 (G6PDd)	*1/*1	gNM	*1/*2	gNM	2	***1/*1B**
**HPQ02**	**G202A/A376G**	5.0 (G6PDd)	***1/*17**	**gRM**	*1/*29	gNM	2	***1/*1B**
**HPQ03**	**G202A/A376G**	3.1 (G6PDd)	***1/*2**	**gIM**	*1/*34	gNM	3	*1/*1
HPQ04	G202A/A376G	2.3 (G6PDd)	*1/*1	gNM	*1/*34	gNM	2	*1/*1
HPQ05	A376G	1.7 (G6PDd)	*1/*1	gNM	*1/*2	gNM	2	*1/*1
HPQ06	C563T	1.8 (G6PDd)	*1/*1	gNM	*1/*34	gNM	2	*1/*1
**HPQ07**	**G202A/A376G**	2.6 (G6PDd)	*1/*1	gNM	***2/*4**	**gIM**	**2**	*1/*1
**HPQ08**	**G202A/A376G**	3.0 (G6PDd)	***1/*2**	**gIM**	*2/*17	gNM	2	*1/*1
HPQ09	G202A/A376G	1.7 (G6PDd)	*1/*1	gNM	*1/*2	gNM	2	*1/*1
**HPQ10**	**A376G**	2.0(G6PDd)	***1/*17**	**gRM**	*1/*17	gNM	2	*1/*1
**HPQ11**	**G202A/A376G**	2.2 (G6PDd)	***1/*17**	**gRM**	***1/*1x3**	**gUM**	**3**	*1/*1
**HPQ12**	**G202A/A376G**	6.0 (G6PDd)	***1/*2**	**gIM**	*1/*2	gNM	1	*1/*1
**HPQ13**	**G202A/A376G**	5.7 (G6PDd)	***1/*2**	**gIM**	*1/*2	gNM	2	*1/*1
**HPQ14**	**Wild type**	7.2 (G6PDn)	***1/*17**	**gRM**	*1/*34	gNM	2	*1/*1
**HPQ15**	**Wild type**	7.8 (G6PDn)	*1/*1	gNM	***1/*4**	**gIM**	**2**	*1/*1
**HPQ16**	**Wild type**	10.5 (G6PDn)	*1/*1	gNM	*1/*34	gNM	1	***1/*1B**
**HPQ17**	**Wild type**	9.7 (G6PDn)	*1/*1	gNM	***4/*17**	**gIM**	**2**	***1/*1B**
**HPQ18**	**Wild type**	8.4 (G6PDn)	*1/*1	gNM	***5/*35**	**gIM**	**1**	***1/*1B**

Abbreviations: G6PDd = G6PD deficient, G6PDn = G6PD normal, G202A/A376G = African A−, A376G = African A+, C563T = Mediterranean, CNV = copy number variation, gNM = normal metabolizer, gIM = intermediate metabolizer, gRM = rapid metabolizer, gUM = ultra-rapid metabolizer, * refers to the star allele.

## Data Availability

Datasets from the current study are available upon reasonable request to the corresponding author.

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
