# Peer review of "Association of CYP2C19, CYP2D6 and CYP3A4 Genetic Variants on Primaquine Hemolysis in G6PD-Deficient Patients"

_pathogens, 2023, doi:10.3390/pathogens12070895_

Round 1

Reviewer 1 Report

The manuscript describe a phamacogenetic study and its association with the G6PD deficiency phenotype aiming better patient management before treatment with 8-aminoquinolines for P. vivax malaria.

The study is well written and of relevance in the field of malaria, nevertheless cautious needs to be taken as the number of patients analyzed might not be representative. The authors decided just to analyze patients with vivax malaria and the presence of hemolysis. An explanation should be given why they decided just to include these patients for the pharmacogenetic study and not consider to compare with vivax patients that did not developed hemolysis. This data could give interesting results of the CYPs and G6PD genotyping that could strengthen the genotype/phenotype correlation found in the patients that developed hemolysis.

Minor comments:

 gNM, gRM and gUM, AKI: need detail description in the text. Only present in the tables legend.

Methods describe the SNP analyzed and informe that was through TaqMan probes, nevertheless does not mention if is custom made or ready made kit. If the last, kit reference should be provided.

Table 3 detail data on copy number variation. This methodology is not described in the methods section.

NA

Author Response

REVIEWER 1

The manuscript describe a pharmacogenetic study and its association with the G6PD deficiency phenotype aiming better patient management before treatment with 8-aminoquinolines for P. vivax malaria. The study is well written and of relevance in the field of malaria, nevertheless cautious needs to be taken as the number of patients analyzed might not be representative.

We thank this reviewer for insightful comments, which have significantly improved the quality of this manuscript, following the specific comments below.

  1. The authors decided just to analyze patients with vivax malaria and the presence of hemolysis. An explanation should be given why they decided just to include these patients for the pharmacogenetic study and not consider to compare with vivax patients that did not developed hemolysis. This data could give interesting results of the CYPs and G6PD genotyping that could strengthen the genotype/phenotype correlation found in the patients that developed hemolysis.

Thanks for the suggestion. The idea was to verify whether G6PD deficiency alone would influence hemolysis or whether CYP2C19, CYP2D6 and CYP3A4 genetic variants would contribute to hemolysis together. Therefore, patients who were hospitalized for hemolysis, with vivax malaria and primaquine treatment were included.

Minor comments:

  1. gNM, gRM and gUM, AKI: need detail description in the text. Only present in the tables legend.

The description of gNM, gRM and gUM was added at the lines 204-207, and 222. The description of AKI was added at the line 55.

  1. Methods describe the SNP analyzed and informe that was through TaqMan probes, nevertheless does not mention if is custom made or ready made kit. If the last, kit reference should be provided.

Was used the kit TaqMan™ Drug Metabolism Genotyping Assay ( ThermoFisher scientific®, following manufacturer instructions. The information was added at the lines 111-113.

  1. Table 3 detail data on copy number variation. This methodology is not described in the methods section.

The CNV methodology was add in the section “2.3. Genotyping of G6PD and CYPs”, at line 115 to 119: “The determination of the number of copies (CNV) of the CYP2D6 gene was performed by real-time duplex PCR, following manufacturer instructions. The reference gene used in the assay was RNAseP. The CNV assay were performed using 7500 Fast Real-Time PCR System, and the amplification product was analyzed using CopyCaller v2.1 software (Applied Biosystems, California, USA).”

Reviewer 2 Report

Influence of CYP2C19, CYP2D6 and CYP3A4 genetic variants on the biological effect of malaria treatment with primaquine in G6PD-deficient and G6PD-normal patients.  By Macedo et al.,

In this work, authors evaluated the association of CYP2C19, CYP2D6 and CYP3A4 variants on the hemolysis in P. vivax patients treated with PQ. Only 18 hospitalized patients (from Manaus, Brazil; 2015-2021) met the inclusion criteria, and whole blood was collected to determine G6PD status and genotyping tests. Clinical and laboratory data were obtained from patients´ records. Among those patients, 13 were G6PDd and the other five were G6PDn. G6PDd individuals showed the worst hemolytic condition during hospitalization. The main result indicated Genotyping of CYP2C19, CYP2D6 and CYP3A4 showed no significant difference in allele frequency between groups (p>0.05).”

In my opinion, the main problem is the limited number of patients to make comparisons and associations and from the results is difficult to infer some effects. I found redundancy among tables and the main text, and too many details in the results.

Title In this work seems that the influence of CYPs on patients with certain conditions was not studied; rather only associations were evaluated.

Abstract

Some phrases need revision e.g. “A series of cases were followed-up at an infectious diseases reference hospital in the Western Brazilian Amazon” might not be accurate since seems that patients were contacted once to take a blood sample only. if not,  please explain. Also, this is not clear why (lines 28-30) “These findings reinforce the importance of studies on the mapping of G6PD deficiency and genetic variations of CYP2C19, CYP2D6 and CYP3A4. This mapping will allow us to validate the prevalence of CYPs and determine their influence on hemolysis” If the results seem to contradict it, “in this study there was no influence of CYPs on hemolysis."

 Material and methods

Page 2. 2.1. study subjects

The text does not mention if the address of patients was or not available (to visit them), as a means to increase the sample. They only mentioned that the telephone number was not available.

Please specify the primaquine treatment scheme and doses, were they given according to their weight or by age group?

Page 3. Authors indicate the following (lines 96-99)“The G6PD status classification was considering for women ≤ 3.9 IU/gHb as deficient, 4.0 – 6.0 IU/gHb as intermediate and ≥ 6.0 IU/gHb normal; for men ≤ 3.9 IU/gHb was considered deficient) and ≥ 4.0 IU/gHb 98 normal [23,24].” However, they allocated participants to two groups “Participants were categorized in two groups: G6PDd and G6PDn (as no intermediate level was separated).” Please indicate the IU ranges for those two groups. Only the median is indicated in the results (lines 148-150) of either group.

Results

please revise the following (line 103) “African A- G202A (rs1050828), African A+ A376G (rs1050829)” versus African A- variant  (G202A/A376G)  vs (lines 181-182)  “the African A- variant 181 (G202A/A376G) was the most common with 55.6%, followed by African A+ (A376G)”

Figure 2. Please be consistent, the title of the graph should be G6PDd, and to compare add a graph for g6PDn.

Table 3. should be more clear to know which patient was G6PDd and G6PDn

Page 4, table 1, needs the meaning of SGOT and SGP in the footnotes.

I suggest revising Table 1, if G6PD deficient and normal patients were separated by these enzyme levels, would be more appropriate to indicate the levels just below the titles (G6PDd (N=13), G6PDn (N=5), not as parameter (Dosage G6PD (UI/g Hb), median (IQR). I also suggest removing the data "total (N=18)" because it takes up space but doesn't add anything else. This is the same for Table 2, as G6PDd and G6PDn groups show everything.

 Lines 156-180, please revise redundancy, everything written in this text in indicated in Table 1. 

Table 3 provides details of each patient; it is not clear if the dosage of G6PD means the level of the enzyme.

For analyzing the relationship of hemolytic aspects and CYP genotypes and phenotypes for each group (G6PDd-table 4 and G6PDn-table 5) were allocated to three CYP levels with different characteristics. I suggest tables 4 and 5 be included as supplementary materials, and in the text only mention the main finding. Those tables do not greatly contribute to the article, and most of the comparisons were not significant.

Although some trends were found between CYP phenotypes and genotypes with G6PD deficiency or with other laboratory parameters, no differences were found between the G6PDd vs. G6PDn groups. Perhaps due to the limited number of patients, so no clear conclusions can be drawn.

Author Response

Influence of CYP2C19, CYP2D6 and CYP3A4 genetic variants on the biological effect of malaria treatment with primaquine in G6PD-deficient and G6PD-normal patients.  By Macedo et al., In this work, authors evaluated the association of CYP2C19CYP2D6 and CYP3A4 variants on the hemolysis in Pvivax patients treated with PQ. Only 18 hospitalized patients (from Manaus, Brazil; 2015-2021) met the inclusion criteria, and whole blood was collected to determine G6PD status and genotyping tests. Clinical and laboratory data were obtained from patients´ records. Among those patients, 13 were G6PDd and the other five were G6PDn. G6PDd individuals showed the worst hemolytic condition during hospitalization. The main result indicated Genotyping of CYP2C19CYP2D6 and CYP3A4 showed no significant difference in allele frequency between groups (p>0.05).”

1. In my opinion, the main problem is the limited number of patients to make comparisons and associations and from the results is difficult to infer some effects. I found redundancy among tables and the main text, and too many details in the results.

We thank this reviewer for insightful comments, which have significantly improved the quality of this manuscript. The text has been completely revised so that all redundancies or doubts are corrected.

2. Title In this work seems that the influence of CYPs on patients with certain conditions was not studied; rather only associations were evaluated.

The title has been changed: “Association of CYP2C19, CYP2D6 and CYP3A4 genetic variants on primaquine hemolysis in G6PD-deficient

3.Abstract: Some phrases need revision e.g. “A series of cases were followed-up at an infectious diseases reference hospital in the Western Brazilian Amazon” might not be accurate since seems that patients were contacted once to take a blood sample only. if not, please explain.

Change was made accordingly: A series of cases were performed at an infectious diseases reference hospital in the Western Brazilian Amazon.

Also, this is not clear why (lines 28-30) “These findings reinforce the importance of studies on the mapping of G6PD deficiency and genetic variations of CYP2C19, CYP2D6 and CYP3A4. This mapping will allow us to validate the prevalence of CYPs and determine their influence on hemolysis” If the results seem to contradict it, “in this study there was no influence of CYPs on hemolysis."

Our results from this article showed that studied deficient individuals with gRM and gUM phenotypes had the highest serum levels of hemolysis markers compared to individuals of normal phenotypes.

4.Material and methods: Page 2. 2.1. study subjects: The text does not mention if the address of patients was or not available (to visit them), as a means to increase the sample. They only mentioned that the telephone number was not available.

In the Amazon region, we have problems with location and difficult access to some neighborhoods and municipalities. In addition, FMT-HVD is a reference hospital for infectious diseases, the population is prejudiced against being approached directly at home or at work. In this region, people change frequently, so the telephone would be the easiest way to be included. In view of this, a previous telephone contact was made to verify the interest and availability if the participants participated in the study, to later include them in the residency.

5.Please specify the primaquine treatment scheme and doses, were they given according to their weight or by age group?

Patients included in the malaria episode received a standard regimen of chloroquine with primaquine, with weight-adjusted dosage. This information was added in line 89-90.

6.Page 3. Authors indicate the following (lines 96-99)“The G6PD status classification was considering for women ≤ 3.9 IU/gHb as deficient, 4.0 – 6.0 IU/gHb as intermediate and ≥ 6.0 IU/gHb normal; for men ≤ 3.9 IU/gHb was considered deficient) and ≥ 4.0 IU/gHb 98 normal [23,24].” However, they allocated participants to two groups “Participants were categorized in two groups: G6PDd and G6PDn (as no intermediate level was separated).” Please indicate the IU ranges for those two groups. Only the median is indicated in the results (lines 148-150) of either group.

The classification used in the work is correct. Table 3 includes the G6PD activity of each participant (UI/gHb). In the study, 3 women were included as intermediate G6PD (5.0 IU/g Hb, 5.7 IU/g Hb e 6.0 IU/g Hb) and all adopted the G202A/A376G variant. Intermediate patients were analyzed along with the deficient ones because they had reduced enzymatic activity.

7. Results please revise the following (line 103) “African A- G202A (rs1050828), African A+ A376G (rs1050829)” versus African A- variant (G202A/A376G) vs (lines 181-182) “the African A- variant 181 (G202A/A376G) was the most common with 55.6%, followed by African A+ (A376G)”

In line 103, in methodology, after the name of the variant, the reference number for these SNPs, which are rs1050828 and rs1050829. In the study, patients with SNPSs G202A and A376G were considered African A-variant, and African A+ only patients with SNP A376G, this information was clarified in line 108 and 109 of the methodology.

8. Figure 2. Please be consistent, the title of the graph should be G6PDd, and to compare add a graph for g6PDn.

The title has been changed: “Frequency of G6PD variants”. Each variant found was included in the legend, being G202A/A376G= African A-; A376G=African A+; C563T=Mediterranean; Wild type =G6PDn.

9. Table 3. should be more clear to know which patient was G6PDd and G6PDn

This information has been added to Table 3.

10. Page 4, table 1, needs the meaning of SGOT and SGP in the footnotes.

The inclusion was made accordingly.

11. I suggest revising Table 1, if G6PD deficient and normal patients were separated by these enzyme levels, would be more appropriate to indicate the levels just below the titles (G6PDd (N=13), G6PDn (N=5), not as parameter (Dosage G6PD (UI/g Hb), median (IQR). I also suggest removing the data "total (N=18)" because it takes up space but doesn't add anything else. This is the same for Table 2, as G6PDd and G6PDn groups show everything.

Information about dose levels for each group has been added, and total values ​​for each group have been removed (tables 1 and 2).

12. Lines 156-180, please revise redundancy, everything written in this text in indicated in Table 1. 

It has been corrected and summarized in the text so that it is not repetitive and redundant.

13. Table 3 provides details of each patient; it is not clear if the dosage of G6PD means the level of the enzyme.

The dosage means G6PD enzymatic activity.

14. For analyzing the relationship of hemolytic aspects and CYP genotypes and phenotypes for each group (G6PDd-table 4 and G6PDn-table 5) were allocated to three CYP levels with different characteristics. I suggest tables 4 and 5 be included as supplementary materials, and in the text only mention the main finding. Those tables do not greatly contribute to the article, and most of the comparisons were not significant.

Tables 4 and 5 were removed from the main text and are now used as supplementary material.

15. Although some trends were found between CYP phenotypes and genotypes with G6PD deficiency or with other laboratory parameters, no differences were found between the G6PDd vs. G6PDn groups. Perhaps due to the limited number of patients, so no clear conclusions can be drawn.

The aim of the study was to know if the patient who had G6PD deficiency together with alterations of Cyps with fast or ultra-fast activity had more hemolysis than who was only dG6PD. Despite the small number included, since the frequency of G6PD deficiency in the Amazon is around 5%, we can observe that the 2 genetic alterations together created a worsening of the hemolytic condition observed in the patients. I added the suggested observations, and all limitations of the study were added.

Reviewer 3 Report

The manuscript titled "Influence of CYP2C19, CYP2D6 and CYP3A4 genetic variants on the biological effect of malaria treatment with primaquine in G6PD-deficient and G6PD-normal patients." by Macêdo M et al., studied G6PD deficiency and genetic variations of Cyp2C19, Cyp2D6 Cyp3A4 genes using standard G6PD assay and real-time PCR, respectively. 

Overall, the manuscript is well written and clearly explains the results. I have made very few minor remarks which could be addressed in the revised version of the manuscript.

1. Please provide the acronyms description at the first appearance of the word such as G6PD, CYP, PQ, G6PDd, G6PDn etc

2. In Table 1 or in methods section, please also include the lowest and he highest ages of the study group. If any children aged less than 5 are involved, are they also treated with chloroquine and primaquine? Please address these two points.

Author Response

The manuscript titled "Influence of CYP2C19, CYP2D6 and CYP3A4 genetic variants on the biological effect of malaria treatment with primaquine in G6PD-deficient and G6PD-normal patients." by Macêdo M et al., studied G6PD deficiency and genetic variations of Cyp2C19, Cyp2D6 Cyp3A4 genes using standard G6PD assay and real-time PCR, respectively.  Overall, the manuscript is well written and clearly explains the results. I have made very few minor remarks which could be addressed in the revised version of the manuscript.

  1. Please provide the acronyms description at the first appearance of the word such as G6PD, CYP, PQ, G6PDd, G6PDn etc

Change was made accordingly.

  1. In Table 1 or in methods section, please also include the lowest and he highest ages of the study group. If any children aged less than 5 are involved, are they also treated with chloroquine and primaquine? Please address these two points.

The minimum age presented by the included participants was 2 years and the maximum age was 60, this information was added in line 157. According to the Brazilian malaria treatment guide, standard treatment was with chloro-quine for 3 days (10 mg/kg on day 1 and 7.5 mg/kg on days 2 and 3) and with primaquine at a dose of 0.5 mg/kg/day for 7 days, including children from 1 year old. For G6PD deficient, the therapeutic regimen contains a weekly dose of primaquine for 8 weeks, adjusted by weight (0.75 mg/kg/day) and should be started on day 4 after treatment with chloro-quine, children from 1 year are treated following the same guideline, this information was added in line 88-93.

Round 2

Reviewer 2 Report

The authors have responded to most of the concerns, only minor revision is now required.

Please add “patients” to the end of the title: “

 Association of CYP2C19, CYP2D6 and CYP3A4 genetic variants on primaquine hemolysis in G6PD-deficient patients.

Page 1. Check spelling “is the most prevalent specie”. Change “specie” to “species”

Page 2. “this study aimed to evaluate the influence of CYP2C19, CYP2D6 and CYP3A4 variants on hemolysis in vivax malaria infected G6PDd patients treated with PQ.”

As indicated earlier the “influence” was not studied, it is rather an association, and please revise it according to the response: “The aim of the study was to know if the patient who had G6PD deficiency together with alterations of Cyps with fast or ultra-fast activity had more hemolysis than who was only dG6PD. “

Page 2. About study subjects, please if possible add some information to understand better the few cases included. Like From your response:  “ the population is prejudiced against being approached directly at home or at work. In this region, people change frequently, so the telephone would be the easiest way to be included

Please revise Figure 2 (page 6) because is confusing, delete from the graph “G6PDd”  as  the chart has both types (G6PDd and G6PDn), then the Wild type is not equal to G6PDn, instead all G6PDn had wild type sequence.

Some spaces are needed, e.g. Table 1, change “(IQR= 2- 3.1) “ to “(IQR = 2 - 3.1). Table 3 “G6PDd = G6PD,” and others. page 8, change “Supplementar”  to “Supplementary“; change “(p>0.05) “ to “(p > 0.05)

5. conclusions. Please clarify or complete “in both G6PD groups.

The manuscript needs a minor editing of English

Author Response

REVISOR 2

Comments and Suggestions for Authors

The authors have responded to most of the concerns, only minor revision is now required.

We thank this reviewer for insightful comments, which have significantly improved the quality of this manuscript, following the specific comments below.

1. Please add “patients” to the end of the title: “Association of CYP2C19, CYP2D6 and CYP3A4 genetic variants on primaquine hemolysis in G6PD-deficient “

The inclusion was made accordingly.

2. Page 1. Check spelling “is the most prevalent specie”. Change “specie” to “species”

The inclusion was made accordingly.

3. Page 2. “this study aimed to evaluate the influence of CYP2C19, CYP2D6 and CYP3A4 variants on hemolysis in vivax malaria infected G6PDd patients treated with PQ.”As indicated earlier the “influence” was not studied, it is rather an association, and please revise it according to the response: “The aim of the study was to know if the patient who had G6PD deficiency together with alterations of Cyps with fast or ultra-fast activity had more hemolysis than who was only dG6PD. “

The purpose of the manuscript was changed according to the title: this study aimed to evaluate association of CYP2C19, CYP2D6 and CYP3A4 genetic variants on primaquine hemolysis in G6PD-deficient patients.

4. Page 2. About study subjects, please if possible add some information to understand better the few cases included. Like From your response:“ the population is prejudiced against being approached directly at home or at work. In this region, people change frequently, so the telephone would be the easiest way to be included”

The address of the participants was not selected to go directly to the residences due to: there is stigma of the population of the hospital of infectious diseases (FMT-HVD) going directly to their residence making it difficult. In addition, the population of endemic areas changes locations a lot, making it difficult for the address available by the health system to be the most appropriate.

5. Please revise Figure 2 (page 6) because is confusing, delete from the graph “G6PDd”  as  the chart has both types (G6PDd and G6PDn), then the Wild type is not equal to G6PDn, instead all G6PDn had wild type sequence.

The inclusion was made accordingly.

6. Some spaces are needed, e.g. Table 1, change “(IQR= 2- 3.1) “ to “(IQR = 2 - 3.1). Table 3 “G6PDd = G6PD,” and others. page 8, change “Supplementar”  to “Supplementary“; change “(p>0.05) “ to “(p > 0.05)“

The inclusion was made accordingly.

7. conclusions. Please clarify or complete “in both G6PD groups.

The inclusion was made accordingly.